# Global certification of wild poliovirus eradication: insights from modelling hard-to-reach subpopulations and confidence about the absence of transmission

Radboud J Duintjer Tebbens, Dominika A Kalkowska, Kimberly M Thompson

Kid Risk, Inc, Columbus, Ohio, USA

**Correspondence to**
Dr Kimberly M Thompson;
kimt@kidrisk.org

## ABSTRACT

**Objective** To explore the extent to which undervaccinated subpopulations may influence the confidence about no circulation of wild poliovirus (WPV) after the last detected case.

**Design and participants** We used a hypothetical model to examine the extent to which the existence of an undervaccinated subpopulation influences the confidence about no WPV circulation after the last detected case as a function of different characteristics of the subpopulation (eg, size, extent of isolation). We also used the hypothetical population model to inform the bounds on the maximum possible time required to reach high confidence about no circulation in a completely isolated and unvaccinated subpopulation starting either at the endemic equilibrium or with a single infection in an entirely susceptible population.

**Results** It may take over 3 years to reach 95% confidence about no circulation for this hypothetical population despite high surveillance sensitivity and high vaccination coverage in the surrounding general population if: (1) ability to detect cases in the undervaccinated subpopulation remains exceedingly small, (2) the undervaccinated subpopulation remains small and highly isolated from the general population and (3) the coverage in the undervaccinated subpopulation remains very close to the minimum needed to eradicate. Fully-isolated hypothetical populations of 4000 people or less cannot sustain endemic transmission for more than 5 years, with at least 20 000 people required for a 50% chance of at least 5 years of sustained transmission in a population without seasonality that starts at the endemic equilibrium. Notably, however, the population size required for persistent transmission increases significantly for realistic populations that include some vaccination and seasonality and/or that do not begin at the endemic equilibrium.

**Conclusions** Significant trade-offs remain inherent in global polio certification decisions, which underscore the need for making and valuing investments to maximise population immunity and surveillance quality in all remaining possible WPV reservoirs.

## BACKGROUND

Achieving the 1988 World Health Assembly polio eradication goal of ending all cases of poliomyelitis[1] requires a successful transition

### Strengths and limitations of this study

► Models the limited but important role of undervaccinated subpopulations in achieving confidence about no wild poliovirus (WPV) transmission after the last reported case.

► Explores trends in transmission and detection for different population sizes as time increases since the last reported case.

► Examines the importance of maximising population immunity and surveillance quality.

► Provides critical information to support decisions related to the ultimate certification of WPV elimination.

► Analyses remain limited by model assumptions, but in abstract provide insights relevant to likely last poliovirus reservoirs.

from the interruption of the current low level of wild poliovirus (WPV) transmission through coordinated cessation of all use of live attenuated oral poliovirus vaccine (OPV) to effective long-term risk management. The Global Polio Laboratory Network supports the Global Polio Eradication Initiative (GPEI) by testing stool samples from acute flaccid paralysis (AFP) cases and sewage samples for polioviruses. As the GPEI approaches success, the transition to the polio endgame has begun. The endgame involves significant complexity, because all countries must achieve and maintain sufficient population immunity[2–4] to stop and prevent the transmission of three separate poliovirus serotypes (ie, 1, 2 and 3) and globally coordinate cessation of each OPV serotype.[5–7] In September 2015, the Global Certification Commission declared successful eradication of serotype 2 WPV (WPV2),[8] which represented a prerequisite to the globally coordinated cessation of all serotype 2-containing OPV use. Global cessation of serotype 2-containing OPV occurred

in late April and early May 2016, during which time over 150 countries stopped using trivalent OPV (tOPV, which contains all three serotypes) and switched to bivalent OPV (bOPV, which contains only serotypes 1 and 3 OPV).[9]

The Global Polio Laboratory Network reported the lowest number of annual paralytic serotype 1 WPV (WPV1) cases in 2017,[10] and no serotype 3 WPV (WPV3) cases since November 2012.[11] Successful WPV eradication requires stopping all transmission, which manifests as an absence of detected WPVs despite actively looking. With increasing time of not seeing cases (while actively looking), confidence increases about WPV die-out. However, the absence of evidence is not evidence of absence. Extended silent transmission can occur, because most poliovirus infections do not lead to symptoms and surveillance gaps can exist. For example, a WPV3 resurfaced in Sudan/Chad in 2004 after no reported cases during 1997–2003[12] and a WPV1 resurfaced in Borno, Nigeria in 2016 after nearly 3 years with no reported cases.[13] The average paralysis-to-infection ratio (PIR), defined as the fraction of infections in fully susceptible individuals that leads to paralytic poliomyelitis (polio) symptoms, equals approximately 1/200, 1/2000 and 1/1000, for serotypes 1, 2 and 3 WPV, respectively.[14] The last reported naturally occurring WPV2 case occurred in India in 1999,[15] and since then, only two episodes of WPV2 infections occurred that traced back to laboratory strains.[16 17] Despite the possibility of silent circulation, the absence of any naturally occurring WPV2 cases for over 15 years (and in many countries for many decades) led to very high confidence about the die-out of WPV2 transmission.

Multiple prior mathematical modelling studies explored the probability of undetected circulation of WPVs in the absence of reported cases or other poliovirus detections. Polio eradication efforts in the Americas, which reported the last indigenous WPV case of any serotype in Peru in 1991,[18] motivated the first analysis and discussion of certification requirements. A statistical analysis of Pan American Health Organization epidemiological data reported less than a 5% chance of undetected indigenous WPV circulation after 4 years since the last reported confirmed case.[19] A simple, stochastic model of poliovirus transmission and die-out characterised the probability of undetected poliovirus circulation in a hypothetical, homogeneously mixed population of 200 000 people in a relatively low-income country, and estimated that not observing a case for 3 years provided 95% confidence about local extinction of WPV infections.[20] This seminal paper provided the foundation for appropriate characterisation of the probability of undetected circulation as a function of the time since the last detected case.[20] Related modelling also explored theoretical thresholds to stop transmission[21] and estimated a minimum population size for persistent transmission of 50 000–100 000 in developing countries and over 200 000 in developed countries required to achieve at least 95% probability of poliovirus persistence for 5 years or more in the absence of vaccination.[22] These studies supported the 2004–2008 GPEI

Strategic Plan requirement of at least 3 years of no polio cases detected by AFP surveillance for certification,[23] which remains the current minimum requirement.[24] A 2012 study[25] relaxed some of the assumptions of the prior theoretical model[20] and highlighted that the probability of undetected circulation varied for different poliovirus serotypes, places and conditions, which suggested the need to focus on appropriate characterisation of conditions in the last likely WPV reservoirs.[25] A 2015 study[26] also used the prior model[20] to show that in the context of an instantaneous introduction of vaccination, the time of the last case relative to vaccine introduction further informs the confidence about the absence of circulation.

Subsequent analyses focused on modelling the conditions in specific and more realistic populations. A 2015 study[27] used a previously developed poliovirus dynamic transmission model[2] applied to: recently endemic transmission in two states in northern India,[28] endemic transmission in northwest Nigeria,[29] a 2010 outbreak in Tajikistan[30] and transmission following a 2013 WPV1 introduction into Israel detected by environmental surveillance.[31] The study characterised the confidence about no undetected poliovirus circulation by serotype as a function of time without reported polio cases or environmental detections considering realistic assumptions for surveillance, immunisation and other national inputs.[27] The results suggested that time periods of 0.5–3 years without detected polio cases provided 95% confidence about the interruption of transmission in the context of perfect AFP surveillance depending on situation-specific characteristics (eg, the overall population immunity, endemic versus outbreak conditions and virus serotype).[27] This model also suggested longer times required for less-than-perfect AFP surveillance and potentially shorter times using highly sensitive environmental surveillance based on the experience in Israel.[27] A recent statistical analysis of the 2013 WPV1 outbreak in Israel demonstrated a rapid increase in confidence about no undetected local transmission following outbreak response immunisation after repeated negative environmental surveillance samples in a city.[32] For Nigeria, a non-dynamic, statistical model[33] estimated a shorter time (compared with the 2015 study[27]) of 14 months required to reach high confidence about no undetected circulation. For its most conservative assumptions about surveillance and force-of-infection, the study estimated a probability of 93% of a WPV-free Africa in the absence of any new WPV cases reported by the end of 2015,[33] shortly before the WPV reemerged.[13] Contrasting with all other modelling studies, a recent study[34] suggested a relatively high probability of undetected circulation after more than 3 years without any polio cases in small populations, although a correction to that analysis emphasised the unrealistic nature of one of the assumptions.[35] Remarkably, the analysis reported that closed populations of 10 000 people or fewer could support many years of transmission in the absence of vaccination and experience gaps between polio cases of over 5 years.[34] A reanalysis of

this hypothetical model identified issues with the analysis and its framing and reported results consistent with the prior literature after correcting for some errors.[36]

Although the modelling results demonstrated the critical importance of sustaining high population immunity through immunisation programmes and high-quality surveillance to obtain high confidence about no undetected circulation, the current GPEI strategic plan only covers 2013–2018,[6] which leads to uncertainty about the ability to sustain high programme performance after 2018. As of mid-2018, questions continue to arise about when the GPEI will cease to exist and what resources will be available to support the polio endgame, including the certification of eradication of WPV1 and WPV3 with high confidence. The GPEI partners already began transition planning, and this process already led to some downsizing of national poliovirus programmes, including the reduction of some AFP surveillance activities.[37] Thus, while the prior modelling assumed strong GPEI and national polio programme performance up through the end of the polio endgame, this assumption now appears optimistic, and further analyses that explore the impact of lower quality surveillance may prove useful in the context of global certification decisions for WPV1 and WPV3 eradication. Further motivation for developing models to support certification decisions comes from the re-appearance of WPV1 in security-compromised areas in Borno, Nigeria after apparent interruption, which raised questions about the ability of poliovirus circulation without detection in communities not (or poorly) accessed by immunisation and surveillance efforts within larger populations with relatively high immunity and good surveillance.

This study aims to support future decisions about WPV certification by: (1) informing confidence about the absence of circulation by modelling the role of hard-to-reach populations, (2) examining the minimum population size required to sustain poliovirus transmission and (3) developing a conceptual framework to provide some structure for future certification decisions.

**Table 1** Model inputs to characterise a hypothetical population that contains an undervaccinated subpopulation

| Model input | Value(s)* | Source/notes |
|---|---|---|
| Total population size | 500 000; **1** million; 5 million | No effect on DEB model behaviour, but required for stochastic analysis of detections |
| Time until vaccination starts, years | | Assumption to characterise hard-to-reach subpopulation within well-vaccinated general population |
| General population | 30 | |
| Undervaccinated subpopulation | 40 | |
| Initial age distribution | | Equilibrium age distribution[38] |
| 0–2 months | 0.01 | |
| 3–59 months | 0.15 | |
| 5–14 years | 0.25 | |
| ≥15 years | 0.59 | |
| Birth rate, births/person/year | 0.02 | 38 |
| Death rate, deaths/person/year | 0.02 | 38 |
| Basic reproduction number ($R_0$) | 10 | 38 |
| Proportion of transmissions via oropharyngeal route | 0.3 | 38 |
| Proportion of contacts reserved for individuals within the same mixing age group | 0.4 | Same value as used in Ref. 38 (not explicitly listed) |
| Average per-dose take rate for serotype 1 OPV | 0.6 | Increased from 0.5 to maintain similar coverage thresholds with different run-up[38] |
| Routine immunisation coverage | | Represents coverage with exactly 3 OPV doses; general population based on Ref. 38, undervaccinated varied around threshold to eradicate, which equals 0.82 for the bolded values in the middle column |
| General population | 0.95 | |
| Undervaccinated subpopulation | 0.75; 0.82; 0.85; 0.90; 0.95† | |
| Proportion of contacts with undervaccinated subpopulation ($p_{within}$) | 0.8; **0.95**; 1.00 | Selected values from Ref. 38 |
| Relative size of the undervaccinated subpopulation compared with total population | 1/20; **1/10**; 1/5 | Selected values from Ref. 38 |
| Paralysis-to-infection ratio | 1/200 | Average for serotype 1 wild poliovirus[2 14] |
| Detection probability per polio case | | Assumption to characterise hard-to-reach subpopulation within general population with high acute flaccid paralysis surveillance quality |
| General population | 0.95 | |
| Undervaccinated subpopulations | 0; 0.1; 0.2; 0.3; 0.4; 0.5; 0.6; 0.7; 0.8; 0.9; 0.95† | |

*Values shown in bold represent values that we held fixed when varying other values in sensitivity analyses.
†All values considered jointly in all sensitivity analysis (hence no single value bolded).
DEB, differential-equation based; OPV, oral poliovirus vaccine.

## METHODS

To inform confidence about the absence of circulation by modelling the role of hard-to-reach populations, we explored the impact of key assumptions using an existing model of a hypothetical population comprised of a well-vaccinated general population and an undervaccinated subpopulation.[38] Table 1 lists the model inputs used to characterise this hypothetical population and explore the role of key assumptions (see online supplementary appendix text, table A1, and online supplementary figures A1, A2 and A3 for model details). We varied several inputs around the base case assumptions indicated by the bold values in table 1, including the degree of mixing between the under vaccinated and general population (pwithin), the relative size of the under vaccinated subpopulation, and the total population size while also assuming a completely isolated under vaccinated population (pwithin=1.0), which implies different absolute sizes of the isolated undervaccinated population. In addition, for each variation around the base case, we simultaneously varied the routine immunisation coverage and detection probability per polio case in the undervaccinated subpopulation. We interpret the total hypothetical population as one epidemiological block (eg, a country) and therefore compute the confidence about no circulation based on all detections that occur in the general population and undervaccinated subpopulation combined. However, we fix the detection probability in the general population at 95% to characterise high-quality national surveillance while considering lower detection probabilities only in the undervaccinated subpopulation (table 1).[38] To estimate the confidence about no circulation in this conceptual model, we use a simplified version (see online supplementary appendix) of the stochastic approach developed by Eichner and Dietz (1996)[20] and adopted by others.[25–27] We define the probability of undetected circulation after a given period of $t$ months without a detection as the number of times in multiple stochastic simulations that $t$ months went by without a detection despite continued circulation, divided by the total number of times that $t$ months went by without a detection (ie, with or without continued circulation). Intuitively, the fraction of all time periods of $t$ months without a detection but with transmission still ongoing should decrease as $t$ increases, corresponding to an increasing probability of no circulation. Confidence about no circulation equals one minus the probability of undetected circulation. To visualise the impact of varying the model inputs, we focus on the time without a detection until the confidence about no circulation first exceeds 95% (CNC95%).

We revisit the question of silent transmission in small populations[22 34 36] using the hypothetical population model[38] in an attempt to inform the bounds on the maximum possible CNC95%. To do so, we ignore the general population and effectively assume a completely isolated and unvaccinated subpopulation and otherwise adopt the hypothetical population assumptions from table 1. We transform the DEB model to a stochastic

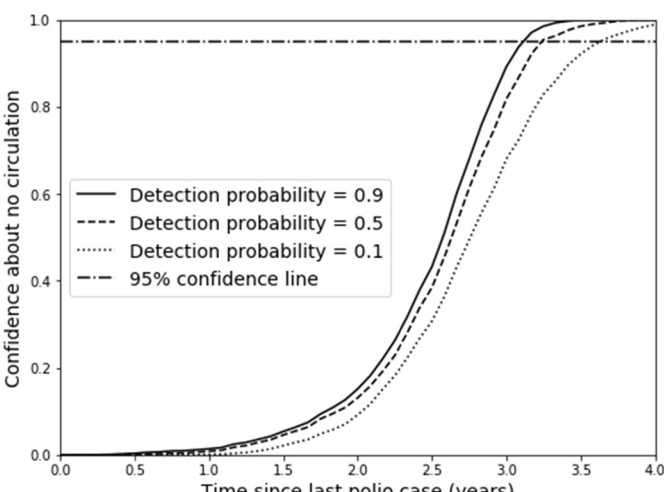

**Figure 1** Confidence about no circulation as a function of time since the last detection for different detection probability values for the hypothetical model base case, with coverage at the corresponding minimum to eliminate WPV (ie, 0.82).

form using the Gillespie algorithm,[39] as described elsewhere,[27] and start either at the endemic equilibrium[34] or with a single infection in an entirely susceptible population. Instead of modelling die-out using the transmission threshold,[2 27] we allow transmission to continue until the infection prevalence becomes 0. This complements the existing work[22 34 36] by providing a comparison to the same situation with a more comprehensive model for poliovirus transmission,[2] adding consideration of the impact of the initial conditions and adding the impact on confidence about no circulation.

Finally, recognising the complexity and inter-related nature of certification decisions, we developed an influence diagram to relate certification timing decisions to outcomes. The diagram provides a conceptual framework to support certification decisions and formulate decisions about the timing of certification as an optimisation problem. The diagram uses conventions from causal loop diagrams[40] and specifies the directionality of relationships between variables using unidirectional arrows. The polarity or sign at the arrow head indicates whether increasing the variable at the base of the arrow increases (+) or decreases (−) the variable that the arrow points to with all else being equal.

### Patient and public involvement

This study did not involve patients or public opportunities for engagement.

### RESULTS

Figure 1 illustrates how the confidence about no circulation increases with time after the last detection as a function of the surveillance quality in the undervaccinated subpopulation (ie, the detection probability). Clearly, a requirement for higher confidence implies the need to wait longer after the last detected case, and lower detection probabilities further increase the time required to

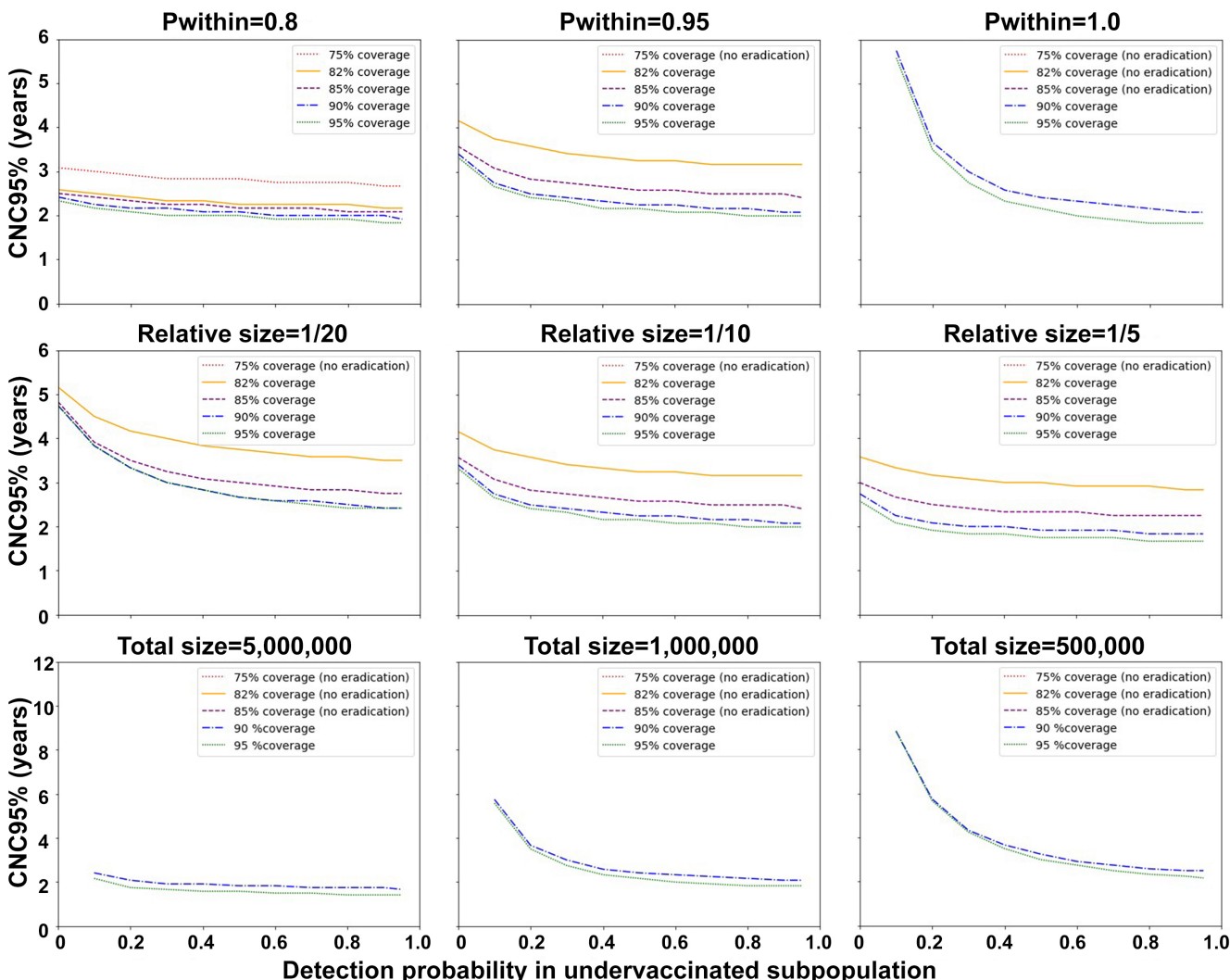

**Figure 2** Time until the confidence about no circulation reaches 95% (CNC95%) from the stochastic analysis for different degrees of isolation of the undervaccinated subpopulation (top row), relative sizes of the undervaccinated subpopulation (middle row) and absolute sizes of a fully-isolated undervaccinated subpopulation that equals 1/10 of the population total (bottom row, note doubled y-axis ranges).

reach a certain level of confidence (eg, the 95% line). Figure 1 shows a relatively modest effect of the detection probability in the undervaccinated subpopulation for this hypothetical model due to continued occurrence of cases in the general population for the assumed degree of mixing (see online supplementary appendix).

Figure 2 shows the CNC95% values (y-axis) as a function of the detection probability for the undervaccinated subpopulation (x-axis) and coverage levels (different curves, see legends). The figure shows longer times required to reach CNC95% values with increasingly more isolated undervaccinated subpopulations (top row, left to right), decreasing relative sizes of the undervaccinated subpopulation (middle row, left to right) and decreasing absolute sizes of the fully-isolated undervaccinated subpopulation modeled by increasing the total size of the population while keeping the relative size of the undervaccinated subpopulation as 1/10 (bottom row, left

to right, note the bottom row uses higher y-axis ranges and assumes pwithin=1.0). The panels in figure 2 omit curves for coverage values that do not result in eradication, because they do not allow for calculation of any confidence about eradication. The panels also omit the data point for 0 detection probability in the event of a fully-isolated undervaccinated subpopulation, because that would imply no ability to detect the virus. Consistent with previous findings,[27] all panels in figure 2 show higher CNC95% values with higher coverage in the undervaccinated subpopulation. In each panel, the lowest shown coverage value may result in the longest period of undetected circulation before interruption and therefore result in the longest time to achieve high confidence about no circulation.

Looking more closely at the differences between the rows, the top row of figure 2 shows a very strong influence of the degree of isolation of the undervaccinated

subpopulation on the CNC95%. With little isolation and no surveillance in the undervaccinated subpopulation, the general population with high surveillance quality can still detect transmission because of relatively frequent spillover of polio cases (see online supplementary appendix). Thus, the results do not depend much on the detection probability in the undervaccinated subpopulation for $P_{within}$=0.8. In contrast, for a fully isolated undervaccinated subpopulation ($P_{within}$=1), the detection probability in this population becomes a more important driver of the CNC95% than the coverage (ie, for detection probability of 0.1 or very poor surveillance and all other inputs at the base case, the CNC95% equals almost 6 years regardless of coverage). The middle row of figure 2 shows CNC95% values of approximately 5 years with no surveillance in a relatively small undervaccinated subpopulation. Although the relative size of the undervaccinated subpopulation affects the mixing dynamics and incidence of cases in both populations, much of the observed effect comes from the implied change in the absolute size of the undervaccinated subpopulation, which directly affects the typical time between cases. As shown in the bottom row of figure 2, changing the absolute size of the undervaccinated subpopulation in the event of full isolation from the general population and a detection probability of 0.1 dramatically affects the CNC95%, which ranges from slightly over 2 years for 500 000 people in the unvaccinated subpopulation (ie 5 000 000 in the total population) to approximately 9 years for 50 000 people in the unvaccinated subpopulation (ie, a 4-fold increase in CNC95% for a 10-fold increase in population size).

Considering the relatively high CNC95% observed for small, isolated populations in figure 2, figure 3A uses a stochastic model to show the distribution of the duration of circulation in a single population not reached by vaccination at all. Figure 3A shows the results as a function of population size for a model initialised at the endemic equilibrium. For very small population sizes (eg, hundreds), not surprisingly poliovirus infections typically die-out within a year, with a maximum duration of circulation of 1 year and 4 months for a closed population of 1000 people (based on 10 000 iterations). The maximum duration of circulation increases rapidly for larger populations. For a population of 5000 people, circulation continues for 3 or more years in 50 of 10 000 (0.5%) iterations. With population sizes of 10 000, 20 000, 30 000, 40 000 and 50 000, circulation continues for at least 10 years for 3%, 34%, 63%, 79% and 88% of iterations, respectively.

Figure 3B shows the same analysis as figure 3A except that it changes the initial conditions by assuming a population with no prior exposure to any polioviruses. In this context, a single introduction rapidly burns through the entire susceptible population and quickly exhausts susceptible individuals, leading to die-out and a maximum duration of circulation of less than 2 years for all population sizes considered in figure 3b. Together, figure 3A and B encompass the bounds on the possible

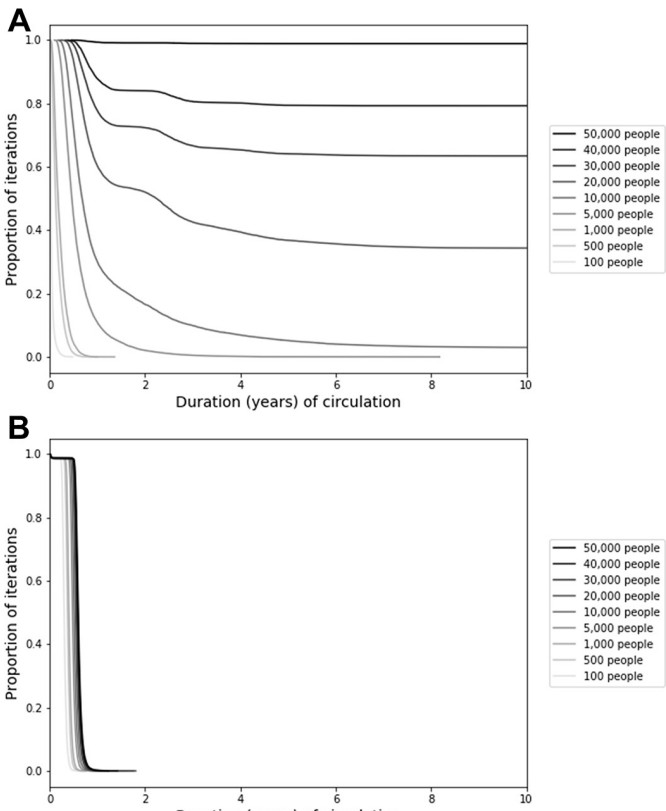

**Figure 3** Results from the analysis of the relationship between population size and persistence of circulation of serotype 1 wild poliovirus transmission in the fully stochastic model when (A) the model starts at the endemic equilibrium and (B) the model starts with a single infection in a fully susceptible population.

duration of circulation for different initial conditions. In reality, small, completely isolated populations are unlikely to remain at the endemic equilibrium because of random fluctuations in the incidence, seasonality and die-out, and no completely naïve populations likely exist. In a separate analysis using the same model, we verified that the addition of seasonality decreases the typical duration of circulation and increases the probability of eradication within 5 years. For example, for a population size of 20 000 people, the probability of eradication within 5 years increased from approximately 64% without seasonality to 78%–92% with a seasonal amplitude of 10% (applied to the basic reproduction number of 10), depending on the timing of the seasonal peak.

While figure 3 implies that increasing the population size results in an increasing probability of persistent circulation (ie, a greater probability of sustained undetected transmission), figure 2 implies that increasing population size decreases the typical time interval between cases (ie, lower probabilities of sustained undetected circulation). Figure 4 shows the net effect of these two opposing trends and suggests that an optimal population size exists around 20 000 people. For smaller population sizes, continued transmission becomes exceedingly unlikely (figure 3), while for larger population sizes, undetected circulation

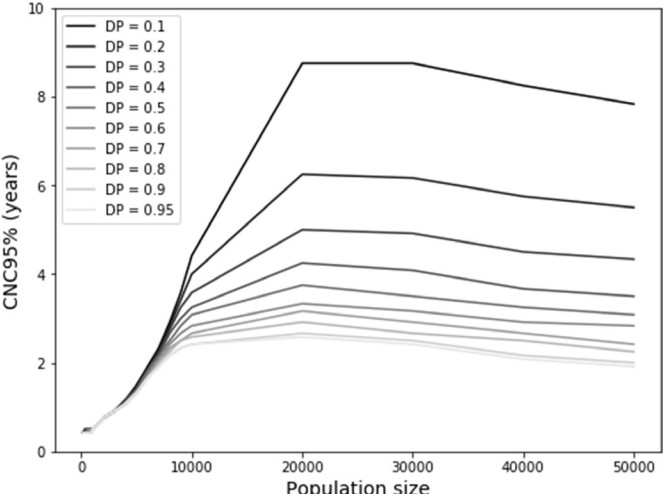

**Figure 4** Time until the confidence about no circulation reaches 95% (CNC95%) for small population sizes in the fully stochastic model that starts at the endemic equilibrium, as a function of DP. DP, detection probability.

becomes less likely due to the higher frequency of cases. This non-linear behaviour suggests a maximum CNC95% of approximately 2.5 years for a detection probability of 1, although the maximum increases to up to 9 years for a very low detection probability of 0.1 and a population size of 20 000–30 000 people.

Figure 5 shows how the desired confidence about no circulation may influence certification timing and key health economic outcomes (see online supplementary appendix text and table A2 for details). Earlier certification and OPV cessation may increase the risk of undetected circulation after OPV cessation (and therefore the possibility of needing to restart OPV use) but may decrease the costs until OPV cessation (and therefore the overall global costs for planned polio immunisation). Therefore, the fundamental optimisation problem consists of finding the desired confidence about no WPV circulation at OPV cessation that minimises the resulting total financial and societal costs. Figure 5 also shows that the costs and risks both depend on the GPEI budget until and after OPV cessation, with a lower budget saving costs in the short term but increasing the time of OPV cessation at a given confidence level and the risks of OPV restarts, which may ultimately result in greater overall costs. Optimisation of the desired confidence about no WPV circulation depends critically on how the confidence about no circulation increases with time after the last detected event from the surveillance system.

## DISCUSSION

Hard-to-reach subpopulations may play a key role in deliberations about WPV circulation and decisions about

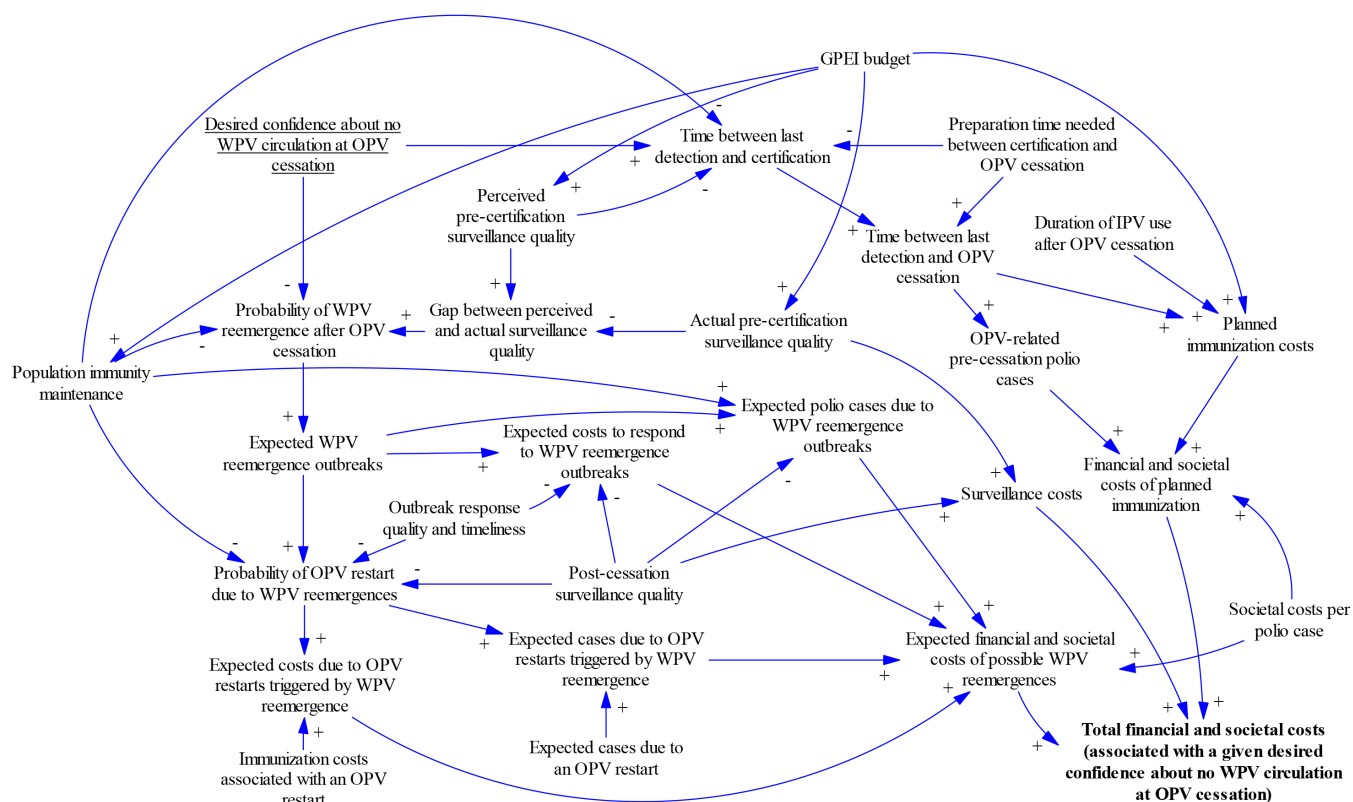

**Figure 5** Conceptual diagram for the implications of choices about the timing of certification of eradication of a WPV serotype on total financial and societal costs. GPEI, Global Polio Eradication Initiative; IPV, inactivated poliovirus vaccine; OPV, oral poliovirus vaccine; WPV, wild poliovirus.

WPV certification. The timing of WPV certification and subsequent OPV cessation involves high stakes and largely depends on the desired confidence about the absence of circulation. Surveillance quality emerges as a key factor that affects both the confidence about the absence of circulation and the ability to detect and control any outbreaks after OPV cessation. However, national surveillance indicators may not suffice to measure the overall surveillance system quality because gaps in surveillance at the level of tens of thousands of people may influence confidence. Our modelling suggests that high-quality surveillance suffices to detect transmission in the context of a relatively well-mixed undervaccinated subpopulation (eg, in Pakistan and Afghanistan),[41] while local gaps may miss transmission for several years in the context of highly isolated undervaccinated subpopulations. With respect to global certification of WPV eradication, this implies a need to address any such gaps in isolated populations that experienced WPV transmission during the last decade. The recent experience in Borno and previously in Chad and Sudan demonstrated the ability of WPVs to circulate undetected for many years in subpopulations missed by both surveillance and immunisation efforts.[12 13] However, one of the main contributions of this work is that it shows that very small, isolated subpopulations cannot sustain transmission indigenously, while in the context of even very limited surveillance, persistent undetected transmission becomes increasingly unlikely for increasing population sizes. To our knowledge, the existence of a worst-case population size for undetected circulation has not yet been demonstrated for polioviruses. Our analysis confirms that with high-quality surveillance, 3 years without a detected WPV case suffices to attain high confidence about no circulation for serotype 1, even considering possible persistence in very small population sizes.

Explicit consideration of the decision to certify WPV eradication (figure 5) suggests that if we remain confident that we can prevent the need to restart OPV due to uncontrolled outbreaks resulting from a possible WPV reemergence, then we should accept a lower confidence about the absence of circulation to certify sooner, because the costs of delaying OPV cessation would outweigh the risk of premature certification. Earlier OPV cessation particularly represents the best option if diminishing GPEI financial and/or global OPV supply resources limit our ability to maintain population immunity and/or respond effectively to post-cessation outbreaks. However, this choice depends on a willingness to accept the reputational risk of finding out that WPV still circulates despite its certification. With WPV3 not detected anywhere since 2012[11] and in many places for decades, the confidence about no WPV3 circulation continues to grow. Although confidence about no circulation increases more slowly for WPV3 than WPV1 due to the lower PIR,[25 27] assuming 1–2 years to prepare for coordinated global OPV cessation, starting the process of removing serotype 3 OPV now would imply at least 7 years of no detection since

the last WPV3 case and synchronised cessation of serotype 3 OPV use (ie, 2012 to 2019–2020). The transition of GPEI resources already occurring leads to expected decreases in population immunity for serotype 3 in some areas. Combined with ongoing serotype 3 vaccine-associated paralytic poliomyelitis, this should motivate careful consideration of the costs, benefits, risks and logistical challenges of globally certifying WPV3 eradication and synchronising serotype 3 OPV cessation before completing WPV1 eradication and serotype 1 OPV cessation, which now appears at least 4 years away.

Our results related to minimum population sizes appear consistent with a prior study[22] that found an average of approximately 5 years of circulation for a population of 20 000 people in a high-$R_0$ setting and an exponential increase in the average duration of circulation with increasing population size. The prior study also reported a higher probability of virus persistence as the degree of mixing between subpopulations increases.[22] Our study suggests that more mixing between subpopulations may not lead to a higher probability of undetected circulation because surveillance can more easily detect persistent viruses for higher degrees of mixing. Using a more realistic model than another prior analysis,[36] we similarly do not find a high probability of persistent transmission for populations of 10 000 people or less.

Like all models, our model makes simplifying assumptions that affect its behaviour.[2] Specifically, we characterised a stylised, hypothetical population to systematically explore key assumptions, used a simplified semistochastic approach to compute CNC95% that does not fully account for all stochastic variability and deterministically characterised die-out. However, for the analysis of small population sizes that depend most on stochastic variability, we accounted for stochastic variability and die-out at the individual level.

While this study highlights the importance of ensuring high surveillance quality in all subpopulations, it also reiterates the role of immunisation in accelerating confidence about no circulation after the last detection.[27] Achieving and maintaining high population immunity to transmission represents a mission critical component of the GPEI.[4] Populations with immunity near the threshold experience increased risk of prolonged undetected transmission. Failing to invest relatively small amounts of resources to maintain high population immunity can lead to much more costly outbreaks, as occurred for example in Tajikistan.[3] Thus, if ensuring high-quality surveillance in all subpopulations remains an elusive goal, then achieving better coverage in those subpopulations would still result in higher confidence about no circulation. In contrast, high-quality surveillance in the context of poor immunisation still leaves the population and the world at risk.

Poliovirus environmental surveillance can detect polioviruses even in the absence of symptomatic polio cases[42 43] and offers the potential to fill some local gaps in symptomatic poliovirus surveillance. For example, the

extensive environmental surveillance system in Israel effectively detected transmission of circulating WPV1 in the absence of any cases and despite very high coverage with inactivated poliovirus vaccine (IPV).[31 44] However, despite the potential for high sensitivity of environmental surveillance to detect infected individuals excreting into the catchment area, its sensitivity remains zero outside of the catchment area and depends on sampling frequency (eg, one sample every year provides little increase in confidence over AFP alone and the quality matters).[45] Environmental surveillance system designs generally depend on access to a centralised sewage network,[43] which hard-to-reach subpopulations (ie, those most likely to sustain undetected poliovirus transmission) may not possess. Further research should help to explore the ability of environmental surveillance to increase confidence about no circulation in specific areas, and the value of the information obtained from environmental surveillance relative to its costs requires evaluation.

IPV plays a relatively limited role with respect to the CNC. While IPV protects otherwise susceptible individuals from paralysis if they become subsequently infected with a live poliovirus and may reduce the participation of these individuals in transmission to some degree, the decreased frequency of paralysis in live poliovirus-infected individuals in the population may delay the detection of any circulating live poliovirus in countries by AFP surveillance (ie, less frequent detection of polio AFP cases depending on IPV coverage). We note the polio AFP detection rate depends on the exposure of fully-susceptible individuals to live poliovirus and it differs from the non-polio AFP detection rate, which the Global Polio Laboratory Network uses to monitor performance of the AFP surveillance system and is not affected by IPV use. Overall, immunisation with IPV helps to maintain population immunity to transmission somewhat, but given births of immunologically naïve, deaths of immune individuals, waning immunity and the absence of circulating live polioviruses, population immunity to transmission declines following WPV eradication and homotypic OPV cessation, even with very high IPV coverage.[46] The extent of transmission possible following reintroduction of a live poliovirus into a country with high IPV coverage will depend on the relative contributions of faecal-oral and oropharyngeal routes to overall transmission.[4] In countries dominated by faecal-oral transmission, the use of IPV will not prevent or stop transmission, and reintroduced live polioviruses that restart transmission may lead to the need to restart the use of OPV.[47]

**Acknowledgements** We thank the Bill and Melinda Gates Foundation for supporting the completion of this work (OPP1129391).

**Contributors** All authors contributed to the study design, model development, interpretation of results, manuscript writing and revisions. RJDT and DAK performed the modelling and analyses and KMT secured the funding for the study.

**Funding** This work was funded by the Bill and Melinda Gates Foundation [OPP1129391].

**Competing interests** None declared.

**Patient consent for publication** Not required.

**Provenance and peer review** Not commissioned; externally peer reviewed.

**Data sharing statement** Technical appendix available on request from the authors.

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
