## [Reviewer comments · BMJ Open]

ARTICLE DETAILS

TITLE (PROVISIONAL)	Global certification of wild poliovirus eradication: Insights from modeling hard-to-reach subpopulations and confidence about the absence of transmission
AUTHORS	Duintjer Tebbens, Radboud; Kalkowska, Dominika; Thompson, KM

VERSION 1 – REVIEW

REVIEWER	Usman Nasir Nakakana MRC Unit at LSHTM, The Gambia
REVIEW RETURNED	26-Jun-2018

GENERAL COMMENTS	There are a lot of assumptions made by the authors on which they based their analysis without substantiating their assumptions with evidence. The outcomes upon which the conclusions are made are not clearly presented, and this makes it difficult to identify in the results. The flow of information in the introduction does not clearly identify the gap and the way that information gap is filled because it seems that the conclusions were already known before the start of the study; this is probably because the outcomes were not clearly stated. The methods are not clear enough to allow a correlation with the results. There is a lot of information but it needs to be presented in a clear and simple language to allow better understanding for even people without any technical knowledge.
--

REVIEWER	Niklas Danielsson Senior Immunization Specialist, UNICEF HQ based in Nairobi
REVIEW RETURNED	23-Aug-2018

GENERAL COMMENTS	I enjoyed reviewing this well-written manuscript. Although admittedly dense for someone like me does not work on modelling I found it rewarding in the understanding of what influences the probability of continued WPV circulation. The manuscript already relates the findings to “real-life” scenarios in Borno, Afganistan and Pakistan but I believe you would increase the interest in the findings among field epidemiologists and vaccination managers if you could widen that discussion further. The Background provides a good summary of current knowledge, the evidence base for the current requirement of a minimum 3 yr of adequate surveillance without detection, and how isolation and population size influence CNC that I found very useful. I miss a discussion on the impact of IPV vaccination on time to CNC95%. IPV protects against paralytic disease and VAPP and shortens the excretion period of Sabin virus after OPV vaccination and therefore impacts on the risks associated with re-introducing OPV
---

	if WPV 1 re-emerges after OPV cessation of OPV. However, there is global shortage of IPV and many countries have had to delay planned introductions of IPV and countries that did introduce a one-dose IPV schedule have had to halt immunization due to vaccine shortage. Your conclusion that the findings support global cessation of OPV3 before OPV1 is important, particularly in view of the persistent circulation of WPV1 in Afghanistan and Pakistan despite huge immunization efforts and the possibility that eradication may take many more years. However, stopping OPV3 globally has large logistic implications that must be weighed against the pragmatic approach of using bOPV “to the end”. The option of early cessation of OPV3 should be seriously considered and carefully evaluated.
--	---

REVIEWER	Jørgen T Lauridsen University of Southern Denmark, Denmark
REVIEW RETURNED	22-Oct-2018

GENERAL COMMENTS	The study explores the extent to which under-vaccinated subpopulations may influence the confidence about no circulation of wild poliovirus (WPV) after the last detected case by using a simulation approach, based on a hypothetical model. Given that the study is professionally performed and lives up to my expectations, and given that it is professionally reported and discussed, I recommend publication without further modifications.
--

VERSION 1 – AUTHOR RESPONSE

Comments from Reviewers:

Reviewer: 1 Comments to the Author

Comment: “There are a lot of assumptions made by the authors on which they based their analysis without substantiating their assumptions with evidence. The outcomes upon which the conclusions are made are not clearly presented, and this makes it difficult to identify in the results. The flow of information in the introduction does not clearly identify the gap and the way that information gap is filled because it seems that the conclusions were already known before the start of the study; this is probably because the outcomes were not clearly stated. The methods are not clear enough to allow a correlation with the results. There is a lot of information but it needs to be presented in a clear and simple language to allow better understanding for even people without any technical knowledge.”

Response: We thank the reviewer for this comment. Throughout the manuscript, we added references to earlier work that provide the details and assumptions. We are trying to find the right balance between explanation and providing a useful manuscript that offers important insights related to the certification of the remaining wild polioviruses. We updated the references that were previously in review, so now an interested reader should be able to get all of the prior foundational work.

Reviewer: 2 Comments to the Author

Comment: “I enjoyed reviewing this well-written manuscript. Although admittedly dense for someone like me does not work on modelling I found it rewarding in the understanding of what influences the probability of continued WPV circulation. The manuscript already relates the findings to “real-life”

scenarios in Borno, Afganistan and Pakistan but I believe you would increase the interest in the findings among field epidemiologists and vaccination managers if you could widen that discussion further.”

Response: We thank the reviewer for appreciating our work and for these comments. We also refer to experience in Chad and Sudan, and in response to this comment, we included additional examples related to Tajikistan and Israel (see discussion).

Comment: “The Background provides a good summary of current knowledge, the evidence base for the current requirement of a minimum 3 yr of adequate surveillance without detection, and how isolation and population size influence CNC that I found very useful. I miss a discussion on the impact of IPV vaccination on time to CNC95%. IPV protects against paralytic disease and VAPP and shortens the excretion period of Sabin virus after OPV vaccination and therefore impacts on the risks associated with re-introducing OPV if WPV 1 re-emerges after OPV cessation of OPV. However, there is global shortage of IPV and many countries have had to delay planned introductions of IPV and countries that did introduce a one-dose IPV schedule have had to halt immunization due to vaccine shortage.”

Response: We thank the reviewer for this comment. We added text to discuss the impacts of IPV vaccination on CNC at the end of the Discussion section.

Comment: “Your conclusion that the findings support global cessation of OPV3 before OPV1 is important, particularly in view of the persistent circulation of WPV1 in Afghanistan and Pakistan despite huge immunization efforts and the possibility that eradication may take many more years. However, stopping OPV3 globally has large logistic implications that must be weighed against the pragmatic approach of using bOPV “to the end”. The option of early cessation of OPV3 should be seriously considered and carefully evaluated.”

Response: We thank the reviewer for recognizing the importance of this work in support of global cessation of OPV3 before OPV1. We agree that logistic implications of such a step, in face of current bOPV using pragmatic approach (including limited mOPV1 stockpile), present a great challenge and we included mention of this.

Reviewer: 3 Comments to the Author

Comment: “The study explores the extent to which under-vaccinated subpopulations may influence the confidence about no circulation of wild poliovirus (WPV) after the last detected case by using a simulation approach, based on a hypothetical model. Given that the study is professionally performed and lives up to my expectations, and given that it is professionally reported and discussed, I recommend publication without further modifications.”

Response: We thank the reviewer for appreciating our work and for this recommendation.

VERSION 2 – REVIEW

REVIEWER	Usman Nasir Nakakana MRC Unit, The Gambia at LSHTM
REVIEW RETURNED	20-Nov-2018c
GENERAL COMMENTS	The language is simpler to understand although the model assumptions and the details of the model are still difficult to

	understand and have not been fully justified. The statements about the role of IPV seem contradictory, if the assumption of effective surveillance holds true, it should not affect the rate of AFP detection, which is in any case, supplemented by environmental surveillance in a setting of high performance of the surveillance system. The role of IPV could be better analysed by comparing different scenarios for better understanding. Some of the assumptions are unrealistic, such as in line 202 onwards, a completely isolated population. Overall, the new information regarding the cessation of vaccination and certification does not appear to be novel, and I don't really see how the findings from this study will significantly affect policy, moving forward.
REVIEWER	Niklas Danielsson UNICEF HQ New York, EPI section, Health Programme Department
REVIEW RETURNED	06-Nov-2018
GENERAL COMMENTS	I am satisfied with the revisions made and congratulate the authors to a very interesting paper.

VERSION 2 – AUTHOR RESPONSE

Comments from Reviewers:

Reviewer: 1 Comments to the Author

Comment: “The language is simpler to understand although the model assumptions and the details of the model are still difficult to understand and have not been fully justified.”

Response: We thank the reviewer for reading the revision and providing further comments, and for noting some improvements. We appreciate that the reviewer would still like additional details about the model, and we would be happy to add more information to the technical appendix. However, we believe that the current information that we provided in the paper and technical appendix provide enough detail for most readers and are sufficient for other modelers to evaluate and replicate the analysis, so we are not sure what to add at this point.

Comment: “The statements about the role of IPV seem contradictory, if the assumption of effective surveillance holds true, it should not affect the rate of AFP detection, which is in any case, supplemented by environmental surveillance in a setting of high performance of the surveillance system. The role of IPV could be better analysed by comparing different scenarios for better understanding.”

Response: We slightly edited the text in the last paragraph to make the role of IPV clearer. IPV protects individual vaccine recipients from paralysis, but it does not significantly change their participation in live virus transmission or population immunity to transmission in countries in which

fecal-oral transmission of live polioviruses matters (i.e., in countries relevant to wild poliovirus certification issues). Thus, it does not matter what scenarios we explore related to IPV use, and we only added text related to IPV to address the misperception that IPV has any significant role with respect to certification in response to a comment from Reviewer 2. Since IPV protects individuals from paralysis, it affects the potential detection of AFP cases associated with circulating wild poliovirus, which is what using AFP for poliovirus surveillance is designed to detect. The reviewer is correct that using IPV does not affect the underlying non-polio AFP rate (i.e., AFP detection), which is a performance indicator for AFP surveillance. As noted in the second-to-last paragraph and demonstrated in Ref. 45, polio environmental surveillance system provides very limited information about most geographies, because environmental surveillance sensitivity is zero in areas that are not covered by sampling (see Ref. 43) and the quality of the existing sites and samples vary such that we do not believe that we should assume that current or future polio environmental surveillance will uniformly promise high quality (the actual quality is now and will likely remain variable).

Comment: "Some of the assumptions are unrealistic, such as in line 202 onwards, a completely isolated population."

Response: This analysis in fact takes on the unrealistic assumptions used in prior publications, which we cite in the paragraph starting on line 202 (line 205 in the tracked version). As we noted "Instead of modeling die-out using the transmission threshold,^{2 27} we allow transmission to continue until the infection prevalence becomes 0. This complements the existing work ^{22 34 36} by providing a comparison to the same situation with a more comprehensive model for poliovirus transmission,² adding consideration of the impact of the initial conditions, and adding the impact on confidence about no circulation." We feel that it is very important to go beyond the current literature to explore the impact of the initial conditions and the impact of small populations on confidence about no circulation based on discussions with GPEI partners, because even more unrealistic prior publications raised questions about the role of small, isolated populations on undetected circulation. We agree with the reviewer about the unrealistic nature of the assumptions in this part, but they reflect the nature of the assumptions made by prior analyses that we seek to address with this part.

Comment: "Overall, the new information regarding the cessation of vaccination and certification does not appear to be novel, and I don't really see how the findings from this study will significantly affect policy, moving forward."

Response: We appreciate the reviewer's perspective, however, this paper adds considerably to the available literature related to the topic of wild poliovirus certification and it addresses questions that Global Polio Eradication Initiative partners have asked us to address.

Reviewer: 2 Comments to the Author

Comment: "I am satisfied with the revisions made and congratulate the authors to a very interesting paper."

Response: We thank the reviewer for reading the revision and for these comments.